# Peripheral Soluble Immune Checkpoint-Related Proteins Were Associated with Survival and Treatment Efficacy of Osteosarcoma Patients, a Cohort Study

**DOI:** 10.3390/cancers16091628

**Published:** 2024-04-24

**Authors:** Binghao Li, Qinchuan Wang, Yihong Luo, Sicong Wang, Sai Pan, Wenting Zhao, Zhaoming Ye, Xifeng Wu

**Affiliations:** 1Department of Orthopedics, Second Affiliated Hospital, Zhejiang University School of Medicine, Hangzhou 310009, China; libinghao@zju.edu.cn (B.L.); yezhaoming@zju.edu.cn (Z.Y.); 2Clinical Research Center of Motor System Disease of Zhejiang Province, Hangzhou 310009, China; 3Department of Surgical Oncology, Affiliated Sir Run Run Shaw Hospital, Zhejiang University School of Medicine, Hangzhou 310016, China; 4Center for Biostatistics, Bioinformatics and Big Data, The Second Affiliated Hospital and School of Public Health, Zhejiang University School of Medicine, Hangzhou 310009, China; yihongluo@zju.edu.cn (Y.L.); wangsicong@zju.edu.cn (S.W.); ps@zju.edu.cn (S.P.); zwt376@zju.edu.cn (W.Z.); 5The Key Laboratory of Intelligent Preventive Medicine of Zhejiang Province, Hangzhou 310058, China

**Keywords:** soluble immune checkpoint-related protein, osteosarcoma, lung metastasis, subtype, survival

## Abstract

**Simple Summary:**

Osteosarcoma is one of the most lethal bone tumors worldwide. Immune checkpoint blockades have achieved significant success in solid tumors; however, their role in osteosarcoma remains obscure. Therefore, we aim to explore the clinical significance of soluble immune checkpoint-related proteins in osteosarcoma in this study. We identified four soluble immune checkpoint-related proteins as predictors of progress-free survival and lung metastasis-free survival of osteosarcoma patients. Our findings indicated that soluble immune checkpoint-related proteins could be promising biomarkers for the outcomes and immunotherapy of osteosarcoma.

**Abstract:**

Background: The immune checkpoint blockade remains obscure in osteosarcoma (OS). We aim to explore the clinical significance of soluble immune checkpoint (ICK)-related proteins in OS. Methods: We profiled 14 soluble ICK-related proteins (BTLA, GITR, HVEM, IDO, LAG-3, PD-1, PD-L1, PD-L2, TIM-3, CD28, CD80, CD137, CD27, and CTLA-4) in the plasma of 76 OS patients and matched controls. We evaluated the associations between the biomarkers and the risk of OS using unconditional multivariate logistic regression. The multivariate Cox model was utilized to develop the prediction model of OS. Immune subtypes were established from the identified biomarkers. Transcriptional data from GEO were analyzed to elucidate potential mechanisms. Results: We found that sTIM3, sCD137, sIDO, and sCTLA4 were significantly correlated with OS risk (all *p* < 0.05). sBTLA, sPDL2, and sCD27 were significantly associated with the risk of lung metastasis, whereas sBTLA and sTIM3 were associated with the risk of disease progression. We also established an immune subtype based on sBTLA, sPD1, sTIM3, and sPDL2. Patients in the sICK-type2 subtype had significantly decreased progression-free survival (PFS) and lung metastasis-free survival (LMFS) than those in the sICK-type1 subtype (log-rank *p* = 2.8 × 10^−2^, 1.7 × 10^−2^, respectively). Interestingly, we found that the trend of LMFS and PFS in the subtypes of corresponding ICK genes’ expression was opposite to the results in the blood (log-rank *p* = 2.6 × 10^−4^, 9.5 × 10^−4^, respectively). Conclusion: Four soluble ICK-related proteins were associated with the survival of OS patients. Soluble ICK-related proteins could be promising biomarkers for the outcomes and immunotherapy of OS patients, though more research is warranted.

## 1. Introduction

Osteosarcoma (OS) is the most common primary bone tumor with peak incidence during adolescence, causing thousands of deaths and disabilities worldwide annually [1,2,3]. Nearly one-quarter of OS patients present with metastatic disease at the time of diagnosis, primarily in the lung. The 5-year survival rate of localized OS patients is approximately 60–70% [4], whereas the rate is less than 20% in metastatic OS patients [5]. Systemic chemotherapy, surgery, and radiation are the mainstay treatments of OS nowadays and have substantially improved the prognosis of OS patients [6,7]. However, the prognosis remains poor for those patients with metastatic or recurrent diseases, and novel treatment approaches and predictive tools are urgently needed [8]. 

Immunotherapy has been revolutionizing the concept of cancer treatment, including osteosarcoma [9]. Chimeric antigen receptor (CAR) therapy, oncolytic virus, and immune checkpoint inhibitors (ICIs) have shown promise for improved osteosarcoma treatment [9]. Immunotherapy-based clinical trials such as SARC028 and PEMBROSARC have been conducted for the efficacy of pembrolizumab (PD-1 inhibitor) in advanced osteosarcoma patients but failed to show positive activity [10,11]. The unfavorable microenvironment for T-cell infiltration and vascular abnormality were the most likely attributed reasons [12]. Identifying patients who would be responsive to ICIs holds promise for the success of this aspect of immunotherapy.

Soluble immune checkpoint-related (ICK-related) proteins are a soluble form of the immune checkpoint receptors/ligands present in the blood [13]. They play a substantial role in the development and treatment of multiple cancers [13,14,15,16]. For instance, soluble CD27 and PD-L2 were identified to be associated with progression and invasive disease prognosis in lung cancer [14]. High blood levels of BTLA and TIM3 correlated with decreased survival in clear-cell renal cell carcinoma [17]. In soft tissue sarcoma, the soluble CD80 level was significantly associated with poor metastasis-free survival in a retrospective study involving 119 patients [18]. There was limited clinical success using immunotherapy for osteosarcoma, partly because of the lack of effective tools to identify patients who would be sensitive to such therapy. Despite a few continuous ongoing clinical trials of ICIs in OS patients, potential associations between soluble immune checkpoint-related proteins and risk or outcomes of OS have not been reported [19]. 

In this study, we systematically profiled 14 soluble ICK-related proteins in a cohort of OS patients and matched healthy controls. Associations between the protein biomarkers and cancer risk, lung metastasis, and progression were identified and validated. Immune subtypes and multi-variable prediction models were established to further explore potential clinical applications in OS patients. 

## 2. Materials and Methods

### 2.1. Study Population and Data Collection

All of the OS patients in this study were recruited from an ongoing bone tumor patient cohort at The Second Affiliated Hospital (SAH), Zhejiang University (Hangzhou, China) initiated in June 2020. This study recruited healthy controls from an ongoing cohort study on healthy individuals at SAH. The protocol of this study was reviewed and approved by the Institutional Review Board of SAH. Written informed consent was signed by all the participants before being enrolled in this study. 

A schematic design of the study is depicted in Appendix A. To identify the role of soluble ICK-related proteins in OS, we implemented a two-stage study. First, we systematically profiled the plasma levels of soluble ICK-related proteins in a cohort of OS patients and matched healthy controls. The biomarkers associated with lung metastasis and progression were identified. The immune subtype based on soluble ICK-related protein levels was also established. Second, we explored the corresponding immune checkpoint gene expression in two external datasets of OS tumors from The Gene Expression Omnibus (GEO), and in silico functional validation of our established immune subtype was performed. 

The patient inclusion criteria of the study were as follows: 1. Pathologically confirmed OS; 2. Not previously treated by surgery or chemotherapy; 3. Informed consent or waiver of consent provided by each participant and follow-up information available. We excluded patients with non-OS or multiple cancers, as well as those who failed to provide informed consent. Detailed clinic-pathological information of the participants was obtained via careful inquiry of the medical record system. Healthy controls and OS patients were matched with gender, age, and the time of enrollment using the propensity matching method (PSM) at the ratio of 1. Epidemiological data were obtained by staff interviewers through in-person interviews using an established study questionnaire at SAH. After the interview and consent, a 20 mL blood sample was collected from each participant into 3 vacutainer tubes (Fisher Scientific, MA), i.e., 2 lavender-top (sodium EDTA) tubes and 1 red (no additive) tube. The plasma samples from the lavender-top tubes were separated and stored in a −80 °C freezer for further study. 

Transcriptomic data for OS were obtained from the datasets GSE99671 and GSE21257 from The Gene Expression Omnibus (GEO, https://www.ncbi.nlm.nih.gov/ (accessed on 15 October 2022)).

### 2.2. Baseline Data Collection, Patient Follow-Up, and Outcomes

Epidemiological information on weight at 3 years before diagnosis (for patients) or recruitment (for controls) was recorded. Body Mass Index (BMI) was calculated by dividing weight by the square of height in meters (kg/m^2^), and it was categorized according to the WHO Asian guideline: underweight or normal weight (<25 kg/m^2^) and overweight or obese (≥25 kg/m^2^). The clinical, pathological, and laboratory test data were retrieved from hospital electronic medical records. Serum LDH and ALP levels at diagnosis (for patients) or at health checkups (for controls) were obtained. OS patients were staged by physicians in charge and pathologists according to the Enneking staging system [20]. The evaluation of response to the neoadjuvant chemotherapy (NAC) of OS patients, which mainly counts for the proportion of necrotic cells in the tumor after NAC, referenced the Huvos classification system [21].

Lung metastasis-free survival (LMFS) was computed using the date of blood collection to the date of the first documented lung metastasis. Progression-free survival (PFS) was calculated from the date of blood collection to the date of first documented recurrence or metastasis or last follow-up or death, whichever came first. Follow-up time was censored at the end of the study or upon patient death, whichever came first. The loss to follow-up was censored in this study. All patients were followed up for survival status until February 2023.

### 2.3. Detection of Soluble ICK-Related Proteins in Plasma

Soluble ICK-related protein levels in plasma were profiled in duplicate using the ProcartaPlex Human Immuno-Oncology Checkpoint Panel (Thermo Fisher, Waltham, MA, USA) in 96-well plate format. Fourteen human immune checkpoint markers (BTLA, GITR, HVEM, IDO, LAG-3, PD-1, TIM-3, CD28, PD-L1, PD-L2, CD80, CD137, CD27, and CTLA-4) were quantified. The assay was performed according to the protocol provided by the manufacturer, applying FLEXMAP 3D (Luminex, Austin, TX, USA) and xPONENT^®^4.3 software. The procedure of protein quantification in plasma had been systematically narrated in our previous study [17]. The biomarkers were stable for detection in plasma within 3 years [22]. The Lower limits of quantification (LLOQ) of the analyte are listed in Appendix A.

### 2.4. Statistical Analysis

Continuous variables were described as the mean ± standard deviation (SD) or median [interquartile range (IQR)], and categorical variables were described as the frequency and percentage. Student’s *t*-test or the Wilcoxon rank-sum test was used to compare continuous variables, and Pearson’s χ^2^-test was used for categorical variables.

The association between each ICK-related protein and the susceptibility of OS was estimated using the unconditional univariable logistic regression and then multivariable logistic regression adjusted by age, gender, and BMI. All soluble ICK-related protein levels were dichotomized based on the median value. The association between each sICK-related protein and the risk of lung metastasis or disease progression was analyzed using the univariate Cox proportional hazards model and then the multivariate Cox proportional hazards model adjusted for age, gender, tumor size, tumor location, LDH, and ALP to identify independent prognostic factors of OS. Cutoff points of ICK-related biomarkers between the high- and low-level groups were calculated using the surv_cutpoint function in the R-package survminer (v0.4.9). For multiple testing, the Benjamin–Hochberg correction method was applied to the *p*-value calculation. R-package survminer (v0.4.9) was used to depict the Kaplan–Meier survival curves, and the differences in LMFS and PFS between different groups were estimated by the two-sided log-rank test.

The determination of immune subtypes of the OS patients was based on the levels of soluble ICK-related proteins using R package ConsensusClusterPlus (v1.62.0) [23]. Principal components analysis (PCA) and t-distributed stochastic neighbor embedding analysis (t-SNE) were performed to show the distribution differences in immune subtypes. Similarly, the immune subtypes were determined according to corresponding ICK-related genes’ expression in dataset GSE21257. The heatmaps were generated employing R-package pheatmap (v1.0.12).

We evaluated the composition of immune cells of different OS patients in dataset GSE21257 using CIBERSORT (CIBERSORT, R script v1.03). The ESTIMATE (Estimation of Stromal and Immune cells in Malignant Tumor tissues using Expression data) score of the OS tumor microenvironment (TME) in the dataset GSE21257 was evaluated using the R-package estimate (v1.0.13) according to the published literature [24]. The association between the expression levels of ICK-related genes and TIICs and TME score were analyzed using Spearman correlation.

The version of R used in the current study was 4.2.2 (×64). All *p* values were two-sided, with values less than 0.05 considered statistically significant.

## 3. Results

### 3.1. Patient Characteristics

A total of 152 participants were enrolled in this study, including 76 OS cases and 76 healthy controls. The demographic and clinical information is listed in Table 1. Among all participants, the mean age of OS cases and healthy controls was 25.6 and 30.4 years old, respectively. Over half of all cases/controls (73.7%) were male, while most OS cases (N = 66, 86.8%) were located in the extremities. There were 1 stage IB, 4 stage IIA, 62 stage IIB, and 9 stage III OS patients according to the Enneking staging system. Moreover, 22 patients had lung metastasis, 8 patients died from the disease, and 19 patients had progressive disease during treatment. Sixty-two patients received neoadjuvant chemotherapy (NAC), where most of the patients (N = 56, 90.3%) had a response to the NAC according to the Huvos classification. The median follow-up period of the OS patients was 13.1 (range: 9.4–21.5) months. 

### 3.2. Soluble ICK-Related Proteins Were Associated with the Susceptibility of OS

The distribution of all 14 soluble ICK-related proteins profiled in this study is listed in Appendix A. Soluble HVEM, GITR, CD28, and PD-L1 were not included in the subsequent analysis because of minimal variation among samples or too many missing values. 

We found that sTIM3 and sCD137 levels were significantly increased in OS patients compared to controls (*p* = 1.86 × 10^−3^ and 4.80 × 10^−4^, respectively), whereas sCTLA4 and sPDL2 levels were significantly decreased in OS patients (*p* = 2.64 × 10^−2^ and 2.73 × 10^−2^, respectively) (Figure 1A). To further validate the results, we analyzed the transcriptomic data of GSE99671 downloaded from the GEO database. *HAVCR2*, *PDCD1LG2*, and *CTLA4* expressions were significantly elevated in tumors compared to normal (*p* = 0.035, 0.024 and 2.0 × 10^−3^, respectively) (Figure 1B).

To further explore the value of soluble ICK-related proteins in the risk evaluation of OS, we performed multivariable logistic regression analysis incorporating age, sex, and BMI. The results indicated that sTIM3, sCD137, sIDO, and sCTLA4 were significantly associated with the OS risk (Table 2). sCTLA4 demonstrated the strongest association with the OS risk (OR = 0.35, 95%CI, 0.15–0.77).

### 3.3. Soluble ICK-Related Proteins Were Associated with Lung Metastasis and Treatment Efficacy of OS Patients

To explore the associations between soluble ICK-related proteins and lung metastasis and cancer progression of OS patients, we performed univariate and multivariate Cox proportional hazard analyses. Univariate Cox analysis showed that sBTLA and sCD80 were significantly associated with decreased risk of lung metastasis, whereas sPDL2 was associated with increased risk of lung metastasis. Also, sBTLA and sPD1 were associated with a decreased risk of progression (Appendix A). In the multivariate Cox proportional hazard analysis, patients with high sBTLA levels had an 85% decreased risk of lung metastasis (HR = 0.15, 95%CI: 0.04–0.60, *p* = 0.007), whereas patients with high sPDL2 levels had an over 13-fold increased risk of lung metastasis (HR = 14.89, 95%CI: 1.61–137.75, *p* = 0.017). sBTLA was also identified to be associated with a decreased risk of cancer progression (HR = 0.25, 95%CI: 0.09–0.76, *p* = 0.014) (Table 3). Kaplan–Meier analysis demonstrated that high sBTLA was associated with increased LMFS and PFS (log-rank *p* = 9.1 × 10^−3^ and 2.2 × 10^−2^, respectively), while high sPDL2 was associated with significantly decreased LMFS and PFS (log-rank *p* = 2.2 × 10^−3^ and 0.05, respectively) (Figure 1C–F).

Furthermore, we found that sTIM3 and sPD1 levels were significantly associated with increased Huvos grade in OS patients who received NAC (All *p* < 0.05) (Figure 1G, Appendix A), indicating that these biomarkers could be used in the clinical prediction of NAC efficacy.

### 3.4. Immune Subtype of OS Based on Soluble ICK-Related Proteins 

The immune subtype of OS was determined using sBTLA, sPD1, sTIM3, and sPDL2 using unsupervised consensus clustering. According to the levels of soluble ICK-related proteins, 35 OS patients were classified as sICK-type1 and 41 patients were classified as sICK-type2, and the consensus matrix (k = 2) is depicted in Figure 2A. All patients could be classified into two clusters according to the results of the t-SNE analysis, which confirmed two different immune subtypes (Figure 2B,C). The distribution of clinic-pathological variables in two subtypes is delineated in the heatmap and Appendix A. Patients belonging to the sICK-type2 subtype had a significantly higher proportion of lung metastasis than patients in the sICK-type1 subtype (Figure 2D, Appendix A). Further multivariate Cox proportional hazard analysis and Kaplan–Meier analysis showed that OS patients with the sICK-type2 subtype had significantly poorer PFS and LMFS than the patients with the sICK-type1 subtype (HR = 3.64, 95%CI 1.06–12.51, log-rank *p* = 2.8 × 10^−2^, and HR = 8.05, 95%CI 1.04–61.96, log-rank *p* =1.7 × 10^−2^, respectively) (Figure 2E,F). 

### 3.5. Functional Exploration of Immune Subtypes in OS

Based on the transcriptomic data of corresponding ICK-related genes identified in tumors from dataset GSE21257, two distinct subtypes were determined using the unsupervised clustering method, which included 27 cases in ICK-type1 and 26 cases in ICK-type2 (Figure 3A). All patients were divided into two clusters according to the results of t-SNE analysis, and this further confirmed two remarkably distinct subtypes (Figure 3B,C). The distribution of clinic-pathological variates in the two subtypes is delineated in the heatmap (Figure 3D, Appendix A). Kaplan–Meier analysis indicated that patients in the ICK-type2 subtype had significantly increased LMFS and PFS compared to ICK-type1 patients (log-rank *p* = 9.5 × 10^−4^, 2.6 × 10^−4^, respectively) (Figure 3E,F), which was opposite to the findings in the blood.

To explore the potential mechanisms underlying our findings, we evaluated the association between ICK-related genes and the immune infiltration of OS tissue from dataset GSE21257. The results indicated that patients in ICK-type2 had significantly higher proportions of M2 macrophages and lower proportions of M0 macrophages than the patients in ICK-type1 (Appendix A). We also generated TME scores using the ESTIMATE algorithm. The results indicated that ICK-type2 cases had significantly higher immune scores, stromal scores, and ESTIMATE scores than those of ICK-type1 (Appendix A). The associations between individual ICK genes and immune cell proportions were also performed and are listed in Appendix A.

## 4. Discussion

In this study, we systematically profiled 10 soluble ICK-related proteins in 76 OS patients and their matched healthy controls. We demonstrated that higher levels of sCD137 and sTIM3 were significantly associated with the risk of OS, and their corresponding genes *TNFRSF9* and *HAVCR2* were significantly elevated in tumor tissues. sBTLA and sPDL2 were significantly correlated with the risk of lung metastasis in OS patients, and sBTLA was also associated with cancer progression. Furthermore, the data on soluble ICK-related proteins allowed us to classify OS patients as sICK-type1 or sICK-type2, which showed significant differences in LMSF. Functional exploration indicated that corresponding ICK-related genes in the tumor could also classify OS patients as one of two immune subtypes. Interestingly, the association between the immune subtype in the tumor and LMSF was opposite to the result from the blood, which indicated the competitive binding of soluble immune checkpoint-related proteins and their corresponding membranous receptors/ligands in the tumor microenvironment (TME) [13]. These results highlighted the prognostic value of these soluble ICK-related proteins in OS patients for the first time, and they also unveiled potential mechanisms in the TME of OS, though more exploration is still warranted. 

We identified sBTLA as the most significant biomarker associated with increased LMSF and PFS in OS patients. BTLA, also known as the B and T lymphocyte attenuator, is one of the most important co-signaling molecules. It is an inhibitory ICK protein that interacts with HVEM and LIGHT on the surface of antigen-presenting cells [25]. BTLA expression was increased in T cells in multiple cancers [26,27], and sBTLA was found to be associated with poor survival in clear-cell renal cell cancer and pancreatic cancer [17,28]. These findings were in contrast to our study. One possibility was that sBTLA could bind to the membrane ligand of BTLA, which would play the role of antagonist of the BTLA/HVEM axis [29], suggesting the distinct role of sBTLA and membranous BTLA in the suppression of T cell activity. Furthermore, an inhibitor targeting the BTLA/HVEM axis has been developed and is now undergoing clinical trials [30], and therefore sBTLA could be a robust player in the treatment of PDAC. Therefore, sBTLA could be an important factor in immunotherapy in OS patients and warrants more research to elucidate the mechanisms.

PD-L2 is another PD1 membranous ligand on the surface of tumor/APC cells that plays a substantial role in the immune evasion of tumors [31]. sPDL2 was associated with an increased risk of lung metastasis of OS in our study. It was also reported to be associated with an increased risk of recurrence in ccRCC [17], platinum resistance and worse outcomes in head and neck squamous cell cancer [32], and the risk of invasive disease in NSCLC [14]. We also found that tumor *PDCD1LG2* expression was associated with improved LMSF, which was opposite to the results in sPDL2. This might be partially attributed to the fact that sPDL2 could derive from the cleavage of membrane-bound PDL2, tumor exosomes, or activated macrophages [33], which could then interfere with the binding of PD1and PDL1/PDL2 in TME, thereby affecting anti-tumor immunity. In other words, PD1–sPDL2 binding could compete with PD1–sPDL1 binding and lead to decreased sPDL1-mediated T-cell cytotoxicity against the tumor, as we observed in our study on the deteriorated survival of OS patients with high levels of sPDL2. Furthermore, we found that *PDCD1LG2* expression was negatively associated with the proportion of monocytes. This is supported by a recent study that demonstrated that monocytes were associated with poor prognosis in osteosarcoma patients [34]. The study partially explains our finding of the association between *PDCCD1LG2* and LMSF. PD-L2 knockdown was found to attenuate the migration and invasion of OS cells via the inhibition of RhoA-ROCK-LIMK2 signaling in vitro [35], which could be another role of PD-L2 in tumor cells rather than in immune cells. Therefore, we propose that sPDL2 is a promising biomarker for predicting lung metastasis of OS. 

CD137 was the first member of the TNF receptor family to be identified as a promising target of immunotherapy [36]. Our results revealed that sCD137 levels were significantly elevated in OS patients, and *TNFRSF9* expression was also significantly elevated in OS tumor tissue. However, no association was identified between sCD137 and the outcomes of OS patients. Our findings could provide some clues in the application of sCD137 in the risk evaluation and treatment of OS. Similarly, we found that sTIM3 was significantly elevated in OS patients compared to healthy donors, while *HAVCR2* expression was also higher in tumor tissues than normal tissues. Cheng et al. reported that TIM3 could mediate the polarization of the M2 macrophage and promote the metastasis of osteosarcoma cells [37], which partially explained our findings, although sTIM3 was not associated with lung metastasis in our study. 

An immune subtype has been established based on immune gene sets in OS using public databases [38]. We successfully constructed the immune subtype based on soluble ICK-related proteins in OS, which demonstrated a distinct classification of the patients and prediction of LMSF. The subtype based on corresponding genes’ expression also explicated identical classification, but the association with LMSF was opposite to their peripheral counterparts. The discrepancy may be derived from the differences in environment between TME and the blood, which is not uncommon and has been reported in lung cancer [39]. The interaction between soluble ICK-related proteins and membrane ICK-related proteins on the surfaces of tumor cells/immune cells could affect anti-tumor immunity and cancer outcomes [13]. 

Our study has several strengths. First, we performed multiplex profiling of soluble ICK-related proteins in both OS cases and healthy controls. Second, we assessed corresponding immune gene expressions in tumors and the association analysis of immune gene expression and immune landscape to provide possible biological mechanisms. Nevertheless, we also acknowledge several limitations. First, we have a limited sample size of patients with mostly stage IIB OS patients, and a relatively short follow-up time, which may restrain the power of our study. To reduce the impact, we implemented multivariate models to adjust our findings, and we also further validated our results by applying online databases. Second, we did not evaluate the level of surface immune checkpoint proteins in circular immune cells (i.e., peripheral blood mononuclear cells), which could be informative to elaborate on the interaction between soluble ICK-related proteins and membrane-bound immune checkpoints on immune cells. Lastly, we did not evaluate the post-treatment level of these soluble ICK-related proteins, which could provide valuable information for the response to treatment in osteosarcoma.

## 5. Conclusions

In this study, we identified four biomarkers associated with PFS and LMFS in OS patients. An immune subtype was established based on our findings. Further exploration and validation were performed using the GEO database. Soluble ICK-related proteins are promising biomarkers for predicting the risk of OS and prognosis in OS patients. Future studies are warranted to validate the four soluble ICK-related proteins in the prediction of prognosis in OS patients and to explore their applications in immunotherapy and other treatment of OS patients. 

## Figures and Tables

**Figure 1 cancers-16-01628-f001:**
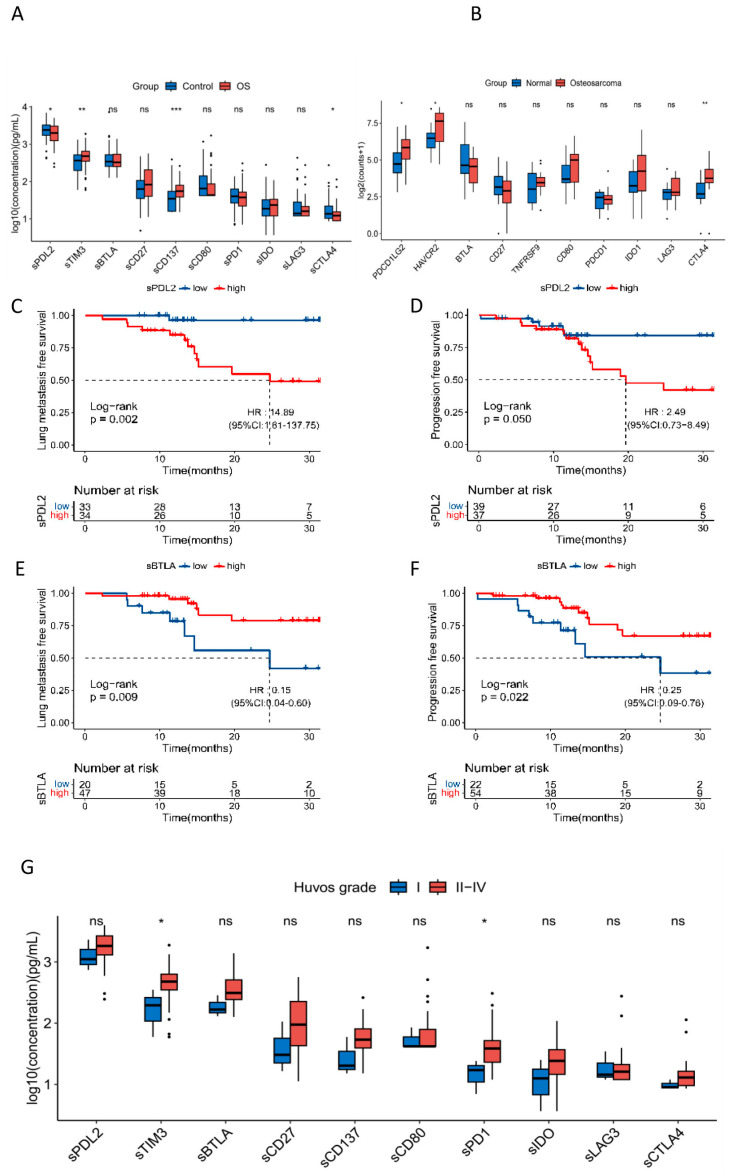
Soluble ICK-related proteins were associated with risk and survival of osteosarcoma. (**A**) Soluble ICK-related proteins were associated with the development of osteosarcoma. OS, osteosarcoma; ns *p* ≥ 0.05, * *p* < 0.05, ** *p* < 0.01, *** *p* < 0.001. (**B**) *HAVCR2*, *PDCD1LG2,* and *CTLA4* expression were significantly elevated in tumors compared to normal among 10 corresponding ICK genes (*p* = 0.035, 0.024 and 2.0 × 10^−3^, respectively) (GSE 99671). * *p* < 0.05, ** *p* < 0.01. (**C**) Patients with high levels of sPDL2 have significantly decreased LMFS compared to the low-sPDL2 patients (log-rank *p* = 2.2 × 10^−3^, HR = 14.89, 95%CI: 1.61–137.75, † *p* = 0.017); (**D**) patients with high levels of sPDL2 have significantly decreased PFS compared to the low-sPDL2 patients (log-rank *p* = 0.05, HR = 2.49, 95%CI: 0.73–8.49, † *p* = 0.145); (**E**) patients with high levels of sBTLA have significantly increased LMFS compared to the low-sBTLA patients (log-rank *p* = 9.1 × 10^−3^, HR = 0.15, 95%CI: 0.04–0.60, † *p* = 0.007); (**F**) patients with high levels of sBTLA have significantly increased PFS compared to the low-sBTLA patients (log-rank *p* = 2.2 × 10^−2^, HR = 0.25, 95%CI: 0.09–0.76, † *p* = 0.014); (**G**) sTIM3 and sPD1 levels were associated with positive response to neoadjuvant chemotherapy in osteosarcoma patients (*p* < 0.05, respectively). ns ≥ 0.05, * *p* < 0.05. † Multi-variable Cox proportional hazard model adjusted with age, gender, tumor size, tumor location, LDH, and ALP.

**Figure 2 cancers-16-01628-f002:**
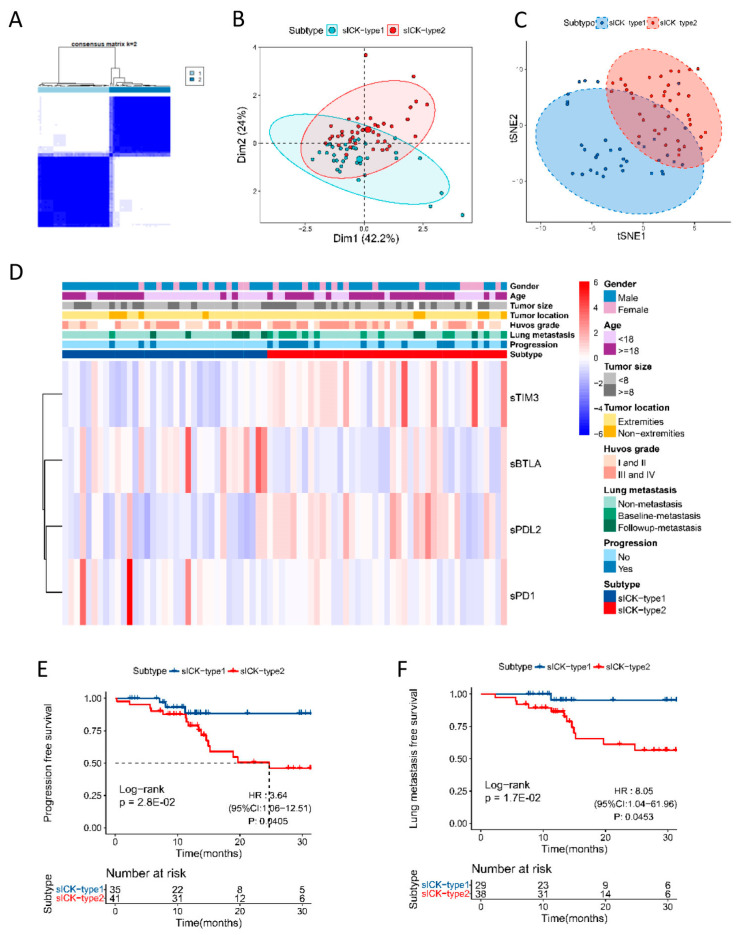
Immune subtypes based on soluble ICK-related proteins in osteosarcoma. (**A**) Consensus clustering matrix for k = 2 based on soluble immune checkpoint-related proteins; (**B**,**C**) principal components analysis (PCA) and T-distributed stochastic neighbor embedding analysis (t-SNE) of different subtypes; (**D**) heatmap of the association between immune subtypes and clinical characteristics. Subtype, gender, age, tumor size, tumor location, Huvos grade, lung metastasis status, and progression status were used as annotations; (**E**) patients in sICK-type2 subtype demonstrated significantly decreased PFS compared to patients in sICK-type1 subtype (HR = 3.64, 95%CI 1.06–12.51, log-rank *p* = 2.8 × 10^−2^); (**F**) patients in sICK-type2 subtype demonstrated significantly decreased LMFS compared to patients in sICK-type1 subtype (HR = 8.05, 95%CI 1.04–61.96, log-rank *p* = 1.7 × 10^−2^).

**Figure 3 cancers-16-01628-f003:**
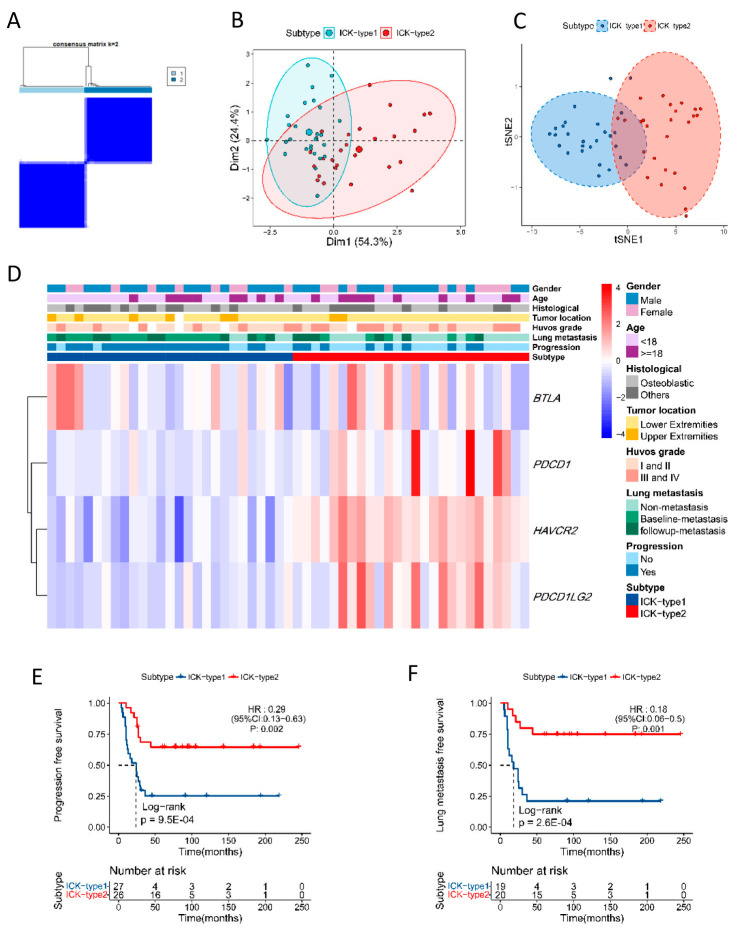
Immune subtypes based on ICK-related genes’ expressions in osteosarcoma. (**A**) Consensus clustering matrix for k = 2 based on immune checkpoint-related genes’ expressions. (**B**,**C**) Principal components analysis (PCA) and T-distributed stochastic neighbor embedding analysis (t-SNE) of different subtypes. (**D**) Heatmap of the association between immune subtypes and clinical characteristics. Subtype, gender, age, histological, tumor location, Huvos grade, lung metastasis status, and progression status were used as annotations. (**E**) Patients in ICK-type2 demonstrated significantly increased PFS compared to patients in ICK-type1 (HR = 0.29, 95%CI 0.13–0.63, log-rank *p* = 9.5 × 10^−4^); (**F**) patients in ICK-type2 demonstrated significantly increased LMFS compared to patients in ICK-type1 (HR = 0.18, 95%CI 0.06–0.50, log-rank *p* = 2.6 × 10^−4^).

**Table 1 cancers-16-01628-t001:** Host characteristics of the participants.

		Osteosarcoma (N = 76)	Health Control (N = 76)	*p*
Age				0.09
	Mean (SD)	25.62 (18.59)	30.41 (15.45)	
Gender				1.00
	Female	20 (26.32)	20 (26.32)	
	Male	56 (73.68)	56 (73.68)	
BMI				0.01
	Mean (SD)	20.90 (4.47)	22.84 (4.90)	
	Missing	0 (0.00)	6 (7.89)	
Tumor location				-
	Extremities	66 (86.84)	-	
	Non-extremities	10 (13.16)	-	
Tumor size				-
	<8 cm	43 (56.58)	-	
	≥8 cm	31 (40.79)	-	
	Missing	2 (2.63)	-	
Enneking stage				-
	IB	1 (1.32)	-	
	IIA	4 (5.26)	-	
	IIB	62 (81.58)	-	
	III	9 (11.84)	-	
Pathological fracture				-
	Yes	13 (17.11)	-	
	No	63 (82.89)	-	
Neoadjuvant chemotherapy				-
	Yes	62 (81.58)	-	
	No	13 (17.11)	-	
	Missing	1 (1.31)	-	
Huvos grade				-
	I	3 (3.95)	-	
	II	21 (27.63)	-	
	III	23 (30.26)	-	
	IV	12 (15.79)	-	
	Missing	17 (22.37)	-	
ALP level at diagnosis				<0.001
	<300 U/mL	55 (72.37)	56 (73.68)	
	≥300 U/mL	20 (26.32)	0 (0.00)	
	Missing	1(1.31)	20 (26.32)	
LDH level at diagnosis				0.00
	≤250 U/mL	54 (71.05)	45 (59.21)	
	>250 U/mL	22 (28.95)	1 (1.32)	
	Missing	0 (0.00)	30 (39.47)	
Lung metastasis				-
	Non-metastasis	54 (71.05)	-	
	Baseline metastasis	9 (11.84)	-	
	Follow-up metastasis	13 (17.11)	-	
Progression				
	Yes	19 (25.00)	-	
	No	57 (75.00)	-	
Dead				-
	Yes	8 (10.53)	-	
	No	68 (89.47)	-	

**Table 2 cancers-16-01628-t002:** Unconditional logistic regression model of markers in cases and controls.

Markers	Univariable Logistic		Multivariable Logistic	
High vs. Low ^a^	OR (95%CI)	*p* Value	Adjusted OR (95%CI) ^b^	*p* Value
sTIM3	**2.11 (1.11–4.06)**	**0.0239**	**2.29 (1.05–5.11)**	**0.0392** ^#^
sCD137	**2.94 (1.53–5.74)**	**0.0013**	**2.72 (1.22–6.25)**	**0.0158** ^#^
sCD27	1.37 (0.73–2.61)	0.3309	1.59 (0.74–3.48)	0.2394
sCTLA4	0.53 (0.28–1.00)	0.0526	**0.35 (0.15–0.77)**	**0.0112**
sIDO	1.23 (0.65–2.34)	0.5166	**2.49 (1.13–5.68)**	**0.0259**
sLAG3	1.53 (0.81–2.91)	0.1952	1.58 (0.73–3.45)	0.2473
sBTLA	0.90 (0.48–1.70)	0.7456	1.10 (0.51–2.39)	0.8113
sCD80	0.66 (0.34–1.24)	0.1952	0.65 (0.30–1.41)	0.2765
sPD1	0.81 (0.43–1.53)	0.5166	0.94 (0.43–2.05)	0.8825
sPDL2	0.73 (0.38–1.38)	0.3309	1.17 (0.53–2.58)	0.7038

Abbreviations: OR Odds ratio, CI Confidence interval. Significant values in bold font; ^a^ High- and low-level groups dichotomized by the median value; ^b^ Adjusted by age, gender, BMI; ^#^ Significant after Benjamini–Hochberg correction for multiple testing.

**Table 3 cancers-16-01628-t003:** Soluble immune checkpoint-related proteins associated with clinical outcomes of osteosarcoma patients.

Markers	Lung Metastasis		Progression	
High vs. Low ^a^	Adjusted HR (95%CI) ^b^	*p* Value	Adjusted HR (95%CI) ^b^	*p* Value
sTIM3	6.58 (0.99–43.9)	0.052	**4.40 (1.16–16.67)**	**0.029**
sCD137	0.60 (0.16–2.31)	0.459	0.88 (0.30–2.55)	0.811
sCD27	**5.73 (1.11–29.67)**	**0.037**	2.61 (0.83–8.21)	0.102
sCTLA4	3.53 (0.89–14.07)	0.074	1.66 (0.53–5.24)	0.385
sIDO	3.34 (0.88–12.64)	0.076	2.46 (0.84–7.17)	0.099
sLAG3	0.54 (0.16–1.85)	0.325	0.68 (0.25–1.83)	0.441
sBTLA	**0.15 (0.04–0.60)**	**0.007** ^#^	**0.25 (0.09–0.76)**	**0.014** ^#^
sCD80	0.28 (0.07–1.23)	0.092	0.50 (0.17–1.46)	0.204
sPD1	0.39 (0.11–1.39)	0.145	0.37 (0.13–1.06)	0.065
sPDL2	**14.89 (1.61–137.75)**	**0.017** ^#^	2.49 (0.73–8.49)	0.145

Abbreviations: HR Hazard ratio, CI Confidence interval. Significant values in bold font; ^a^ High and low groups dichotomized by the R-package survminer; ^b^ Adjusted by age, gender, tumor size, tumor location, LDH, and ALP. ^#^ Significant after Benjamini–Hochberg correction for multiple testing.

## Data Availability

The datasets used and analyzed during the current study are available upon reasonable request.

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
