# Peer review of "Peripheral Soluble Immune Checkpoint-Related Proteins Were Associated with Survival and Treatment Efficacy of Osteosarcoma Patients, a Cohort Study"

_cancers, 2024, doi:10.3390/cancers16091628_

Round 1

Reviewer 1 Report

Comments and Suggestions for Authors

The study is very interesting and may provide good information for the reader of the journal. However, The four detected biomarkers need to be further  analyzed in patients ‘samples before and after the treatment has been initiated to assess their clinical utilization.

Many thanks

Author Response

We thank the reviewer for the appreciation for our study. The reviewer suggested to further analyze the four detected biomarkers in patients’ samples before and after the treatment to assess their clinical utilization. We agree with the reviewers suggestion and we believe that the assessment could firmly explore the detected biomarkers’ application in the clinic. Unfortunately, we only collected the pretreatment samples in our protocol, currently we do not have the post-treatment sample. Alternatively, we search the related literatures in the PubMed. A recently published study indicated that post-treatment sPD-L1 level was significantly elevated compare to pre-treatment level in 128 patients with advanced stage solid tumor who received immune checkpoint blockade (ICB) therapy. However, pre-treatment sPD-L1 was still considered as independent predictive biomarker of OS and PFS [1]. Another study profiled the proteomics in both the pre-treatment (ICB) and on-treatment plasma employing the Somalogic aptamer-based assay in 225 NSCLC patients. The results showed that sPD-1 significantly elevated in on-treatment samples [2]. However, no study reported difference in circulating proteins between pre-surgery and post-surgery in osteosarcoma patients. Therefore, we listed this point as one of our study limitations in the discussion part.

  1. Oh, S.Y.; Kim, S.; Keam, B.; Kim, T.M.; Kim, D.W.; Heo, D.S. Soluble PD-L1 is a predictive and prognostic biomarker in advanced cancer patients who receive immune checkpoint blockade treatment. Sci Rep 2021, 11, 19712, doi:10.1038/s41598-021-99311-y.
  2. Bar, J.; Leibowitz, R.; Reinmuth, N.; Ammendola, A.; Jacob, E.; Moskovitz, M.; Levy-Barda, A.; Lotem, M.; Katsenelson, R.; Agbarya, A.; et al. Biological insights from plasma proteomics of non-small cell lung cancer patients treated with immunotherapy. Front Immunol 2024, 15, 1364473, doi:10.3389/fimmu.2024.1364473.

Reviewer 2 Report

Comments and Suggestions for Authors

Before acceptance, a few points should be addressed:

1. To cope with the complexity of the study, the readability and flow of thoughts can be improved, e.g. in the Discussion (l.345-346: "......infiltrating immune cells (TIICs) analysis [35], partially explaining.........", l.356-358: "Nonetheless, we believe that sCD137 could be an active player in the risk evaluation and treatment of OS, as one of the most promising immune checkpoint proteins for cancer immunotherapy in TME [37]."

2. Some required details of warranted future studies should be included in the Conclusions section.

Comments on the Quality of English Language

A careful editing and some rewriting as indicated in the report will be helpful.

Author Response

Before acceptance, a few points should be addressed:

  1. To cope with the complexity of the study, the readability and flow of thoughts can be improved, e.g. in the Discussion (l.345-346: "......infiltrating immune cells (TIICs) analysis [35], partially explaining.........", l.356-358: "Nonetheless, we believe that sCD137 could be an active player in the risk evaluation and treatment of OS, as one of the most promising immune checkpoint proteins for cancer immunotherapy in TME [37]."

Response: We thank the reviewer for the precious comments. We carefully revised the manuscript and we hope the readability and flow of the thoughts are improved according to the reviewer.

  1. Some required details of warranted future studies should be included in the Conclusions section.

Response: The details of the warranted future studies has been added to the Conclusions.

Reviewer 3 Report

Comments and Suggestions for Authors

In the manuscript by Li et al. titled, "Peripheral soluble immune checkpoint-related proteins were associated with survival and treatment efficacy of osteosarcoma patients, a cohort study," the authors share their findings about the potential of using soluble immune checkpoint proteins as biomarkers for patients with osteosarcoma. The cohort comparison size is appropriate for such a heterogenous cancer. The introduction is well written and is appropriate for understanding the later results. The results section is also well written; however, the presentation of data needs to be improved. One needed clarification is interchangeable abbreviations of osteosarcoma and overall survival. Both are OS, but only one should be used and the other should be spelled out. 

Figure 1

Quality of images is not high resolution. 

They note that three soluble proteins were not included due to poor results, but 1A and 1F only have 10 proteins. Also it would be best to include corresponding RNA data for each protein from 1A in 1B. 

The survival curves in 1C-1E are too small. Also, the authors should include the progression free survival for sPDL2 high and low. 

Table 2

It would be preferable to do bar graphs of the CI of the OR. At the very least, to make it easier to see the data, all the "1 (reference)" can be removed and explained in the text or legend. This is needed for all tables in results and supplemental data.

In the text it mentions 3 significant soluble proteins that were significant in the multivariable analysis; however, they omit sIDO.  

Table 3

In the text, it states that high sBTLA is associated with increased LMFS  and PFS; however, the adjusted HR values are 0.15 and 0.25. This suggests that there is a decreased risk of lung metastasis and disease progression. Similarly, the reference to sPDL2 having a HR of 14.89 should be associated with an increased risk in lung metastasis. Also the p values provided are not the same in the text as they are in the table. 

Figure 2

The identification of 2 subsets of osteosarcoma based on soluble proteins is not as "remarkably" distinct as the authors state. The ambiguity of the two subtypes is further highlighted in Figure 2D and it appears that the subsets have equal representation of patient demographic factors in both. However, Kaplan-Meyer analysis of progression free survival and lung metastasis free survival are significantly different and favor type 1. 

Figure 3

The distinction between type 1 and type 2 was more apparent using the transcriptomic data, including the type 1 decreased expression of 4 mRNA correlating with patient demographic information of progression and lung metastasis. The Kaplan-Meir analysis showed this favorable association for type 2. This comes as a surprise that the mRNA and protein data would have conflicting data. 

Discussion

The authors did not systematically profile 14 soluble proteins, they only validated 10. 

Comments on the Quality of English Language

There were several instances where the past tense of the verb was incorrectly used. 

Author Response

In the manuscript by Li et al. titled, "Peripheral soluble immune checkpoint-related proteins were associated with survival and treatment efficacy of osteosarcoma patients, a cohort study," the authors share their findings about the potential of using soluble immune checkpoint proteins as biomarkers for patients with osteosarcoma. The cohort comparison size is appropriate for such a heterogenous cancer. The introduction is well written and is appropriate for understanding the later results. The results section is also well written; however, the presentation of data needs to be improved. One needed clarification is interchangeable abbreviations of osteosarcoma and overall survival. Both are OS, but only one should be used and the other should be spelled out. 

Response: We thank the reviewer for the appreciation of our study. We agree with the reviewer that it is easy to misunderstand for the abbreviations of osteosarcoma and overall survival. However, the endpoints in our study are Lung metastasis free survival (LMFS) and Progression free survival (PFS). Overall survival is not used as endpoint in this study, the OS is only used for the abbreviation of osteosarcoma.

Figure 1

Quality of images is not high resolution. 

They note that three soluble proteins were not included due to poor results, but 1A and 1F only have 10 proteins. Also it would be best to include corresponding RNA data for each protein from 1A in 1B. 

The survival curves in 1C-1E are too small. Also, the authors should include the progression free survival for sPDL2 high and low. 

Response: We thank the reviewer for the comments. The Figure was revised according to the reviewer’s comments.

Table 2

It would be preferable to do bar graphs of the CI of the OR. At the very least, to make it easier to see the data, all the "1 (reference)" can be removed and explained in the text or legend. This is needed for all tables in results and supplemental data.

In the text it mentions 3 significant soluble proteins that were significant in the multivariable analysis; however, they omit sIDO.  

Response: The table 2 have been revised as the reviewer suggested.

Table 3

In the text, it states that high sBTLA is associated with increased LMFS  and PFS; however, the adjusted HR values are 0.15 and 0.25. This suggests that there is a decreased risk of lung metastasis and disease progression. Similarly, the reference to sPDL2 having a HR of 14.89 should be associated with an increased risk in lung metastasis. Also the p values provided are not the same in the text as they are in the table. 

Response: We thank the reviewer for the comments. The text has been revised.

Figure 2

The identification of 2 subsets of osteosarcoma based on soluble proteins is not as "remarkably" distinct as the authors state. The ambiguity of the two subtypes is further highlighted in Figure 2D and it appears that the subsets have equal representation of patient demographic factors in both. However, Kaplan-Meyer analysis of progression free survival and lung metastasis free survival are significantly different and favor type 1. 

Response: We are sorry for the inappropriate description. The text has been revised to fit the results.

Figure 3

The distinction between type 1 and type 2 was more apparent using the transcriptomic data, including the type 1 decreased expression of 4 mRNA correlating with patient demographic information of progression and lung metastasis. The Kaplan-Meir analysis showed this favorable association for type 2. This comes as a surprise that the mRNA and protein data would have conflicting data. 

Response: We agree with the reviewer. Part of our finding is the opposite findings between mRNA in tumor and protein in blood, which may indicate the interactions of immune checkpoints in TME.

Discussion

The authors did not systematically profile 14 soluble proteins, they only validated 10. 

 Response: We thank the reviewer for pointing this out. The number of soluble proteins has been revised in the manuscript.
